# Deep Reinforcement Learning-Based Video Offloading and Resource Allocation in NOMA-Enabled Networks

Siyu Gao [1,2], Yuchen Wang [1,2], Nan Feng [1,2], Zhongcheng Wei [1,2] and Jijun Zhao [1,2,*]

1   School of Information and Electrical Engineering, Hebei University of Engineering, Handan 056038, China; yayaye1216@163.com (S.G.); ssdlhy123450@163.com (Y.W.); fengnan3128@163.com (N.F.); weizhongcheng@hebeu.edu.cn (Z.W.)
2   Hebei Key Laboratory of Security and Protection Information Sensing and Processing, Hebei University of Engineering, Handan 056038, China
*   Correspondence: zjijun@hebeu.edu.cn

**Abstract:** With the proliferation of video surveillance system deployment and related applications, real-time video analysis is very critical to achieving intelligent monitoring, autonomous driving, etc. Analyzing video stream with high accuracy and low latency through the traditional cloud computing represents a non-trivial problem. In this paper, we propose a non-orthogonal multiple access (NOMA)-based edge real-time video analysis framework with one edge server (ES) and multiple user equipments (UEs). A cost minimization problem composed of delay, energy and accuracy is formulated to improve the quality of experience (QoE) of the UEs. In order to efficiently solve this problem, we propose the joint video frame resolution scaling, task offloading, and resource allocation algorithm based on the Deep Q-Learning Network (JVFRS-TO-RA-DQN), which effectively overcomes the sparsity of the single-layer reward function and accelerates the training convergence speed. JVFRS-TO-RA-DQN consists of two DQN networks to reduce the curse of dimensionality, which, respectively, select the offloading and resource allocation action, as well as the resolution scaling action. The experimental results show that JVFRS-TO-RA-DQN can effectively reduce the cost of edge computing and has better performance in terms of convergence compared to other baseline schemes.

**Keywords:** mobile edge computing (MEC); non-orthogonal multiple access (NOMA); video offloading; resource allocation; deep reinforcement learning (DRL)





## 1. Introduction

Along with the development of the communication infrastructure and embedded systems, a number of cameras have been deployed around the world to collect environmental information, including traffic monitoring, electronic health care, object tracking, and smart robotics [1]. According to Cisco's forecast, video streaming will account for 80% of total Internet traffic by 2023 [2]. In particular, surveillance cameras can produce video transmissions of nearly 25–30 frames per second. Low frame rate (1.25 Hz) moving image sequences generate more than 100 Mb data per second. However, camera sensors have limited computing ability and only support low-complexity recognition algorithms, which means that the video recognition accuracy is limited. In addition, deploying a network which only uses cameras to meet computing requirements is too costly for the system. To obtain the information from a video, it is necessary to send the video frames taken by the user equipments (UEs) to a data center with abundant calculation resources, and deal with the desired scene information. However, the bandwidth, which is needed to efficiently transmit, and the analysis accuracy of the video stream is prohibitive. In addition, video analytics are also computation-intensive, and analysis of the video on the UEs and the cloud data centers alone cannot meet the requirements of resource and delay. Although many researchers have tried to solve this problem, the huge challenge of the video analysis

delay is due to a lack of effective performance. Mobile edge computing (MEC)-based video analysis is the only feasible method to satisfy the demand for a large volume of video streams in real-time.

As an emerging computing architecture, MEC can decentralize the computing ability from centralized cloud centers to users and mobile devices, which are close to the edge of the network [3]. Computing the task at the edge of the network can not only reduce the delay and the load of bandwidth, but also reduce the risk of privacy leakage and improve data security. Most of the existing articles on task offloading adopt an orthogonal frequency division multiple access technology (OFDMA) [4], which can only provide a single channel resource for the UEs and has low bandwidth resource utilization. To improve the utilization of bandwidth resources, non-orthogonal multiple access (NOMA) has been proposed [5]. Unlike orthogonal multiple access technology (OMA), NOMA can accommodate more users via non-orthogonal resource allocation, that is, it can provide services to multiple users on the same subchannel at the same frequency and time, thus improving spectral efficiency [6,7]. Applying NOMA technology to the video offloading process can effectively increase the capacity of the bandwidth resources, reduce the delays of video stream offloading, and improve the user's quality of experience (QoE) [8].

However, real-time video analysis is not easy. Offloading video to the edge for analysis requires high demands on resources and latency, which needs to dynamically balance the transmission and computation processes. Specifically, video analysis usually has a high demand for resources. Analyzing the video frame by frame requires a large amount of computing resources [9,10], which may easily lead to a long delay [11]. Moreover, the complex network structure necessitates the consumption of several GBs of memory. Due to the limitation of the computational resources of the edge, it is necessary to enact a reasonable offloading decision and resource allocation to satisfy the demand of delay. Generally, the offloading decision optimization problem is always combined with resource allocation, which leads to a non-convex NP problem, where it is difficult to determine the best way to allocate resources in a distributed environment.

Recently, deep reinforcement learning (DRL) has been widely used in many applications of mobile communication [12]. Using DRL to improve the performance of large-scale dynamic video analysis is quite interesting, however, the video analytic system lacks an effective mechanism to optimize the video configuration adaptively, which causes low resource utilization. Specifically, we use the video frame resolution as an example to illustrate the impact of the video configuration on the accuracy and latency of video analysis. In this paper, we divide the total time into several equal time slots and define "fps" as the numbers of video frames for each time slot. Images with a higher resolution may be more accurate, however, they may cause a long transmission delay. When the network bandwidth is time-varying, translating video with a higher resolution may increase resource consumption and cause high latency, while translating video with a lower resolution may reduce the network utilization and the analysis accuracy. Therefore, it is necessary to propose an algorithm to select the video frame resolution adaptively, according to the state of the network.

Based on the above discussion, this paper proposes the joint video frame resolution scaling, task offloading, and resource allocation algorithm based on Deep Q-Learning Network (JVFRS-TO-RA-DQN) to optimize video edge offloading and the resource allocation decision. The key contributions of this paper are as follows.

- A two-layer NOMA-enabled video edge scheduling architecture is proposed, where UEs are divided into different clusters of NOMA, and the tasks generated by UEs in the same cluster are offloaded over a common subchannel to improve the offloading efficiency.
- An attempt is made to optimize the QoE of the UE by formulating a cost-minimization problem composed of delay, energy, and accuracy in order to weigh up the relationship between these three parameters.
- The JVFRS-TO-RA-DQN algorithm is proposed to solve the joint optimization problem. The JVFRS-TO-RA-DQN algorithm contains two DQN networks; one is used to select offloading and resource allocation action, and the other is used to select video frame

resolution scaling action, which effectively overcomes the sparsity of the single-layer reward function and accelerates the training convergence speed.
- The experimental results show that the JVFRS-TO-RA-DQN algorithm can achieve better performance gains in terms of improving video analysis accuracy, reducing total delay, and decreasing energy consumption compared to the other baseline schemes.

The rest of this article is organized as follows. In Section 2, we review the relevant work carried out in other studies. Section 3 provides a description of the problem and the scheduling model, and Section 4 depicts the details of our algorithm's implementation. In Section 5, the results of the simulation are analyzed, and finally, we summarize our work in Section 6.

## 2. Related Works

### 2.1. NOMA-Enabled Task Offloading in MEC Scenarios

Video surveillance systems have been used extensively in various industries, and are gradually becoming intelligent, for example, face recognition and object detection [13,14], etc. Along with the rapidly growing number of video monitoring applications, it is necessary to transfer and analyze an increasing amount of video data. MEC has great potential in terms of reducing delays [15–18], and energy consumption [19–21] due to its intelligence in computing and caching. It is possible to reduce the transmission burden and latency by offloading the computation-intensive and latency-sensitive tasks to the MEC server. In order to further optimize the spectrum resource allocation and achieve high speed transmission and wide coverage, most studies combined NOMA and MEC. The authors in [22] focused on the partial offloading and binary offloading problems under time division multiple access and NOMA and tried to maximize the computing efficiency of the system. The authors in [23] proposed the ultra-dense heterogeneous network (UDHN) based NOMA-MEC system and studied the resource allocation problem of multi-SBS and multi-users to minimize user energy consumption and task delay. The authors in [24] focused on reducing the transmission delay and optimizing workload offloading allocation in a downlink NOMA-based MEC system. To solve this problem, they designed a channel quality ranking algorithm to obtain the optimal offloading decision. The authors in [25] considered the random task arrival and the uncertainty of channel conditions, and they proposed a decentralized DRL framework to solve the problem of power allocation, where the state was based on local observations. In addition, NOMA technology is also applied in many practical situations to improve spectrum utilization, such as robotics, unmanned aerial vehicle (UAV), and smart healthcare scenarios, e.g., where multiple users offload tasks to the MEC simultaneously. The authors in [26] proposed a communication enabled indoor intelligent robots (IRs) service framework, which adopted the NOMA to support the highly reliable communications. The efficiency and communication reliability of the IRs was maximized using a DRL-based algorithm. The authors in [27] considered a framework for computation offloading in which UAVs used NOMA and MEC techniques to serve mobile users. They introduced federated learning and reinforcement learning to solve the problem of privacy restriction between the UAVs. In order to satisfy the ultra-reliable low-latency connectivity requirements of the remote-e-Health systems, the authors in [28] considered applying a NOMA to the e-Health systems and proved that the NOMA exhibits an excellent performance in the scenarios of fifth generation and beyond. These solutions provide certain insights in to applying a NOMA to enable efficient task offloading in MEC scenarios, while also providing a feasible scheme through which to solve the efficient transmission of video data.

### 2.2. Video Analysis in MEC Scenarios

Due to the increasing demand of video surveillance applications for resources [29], offloading them to edge devices for computing has been widely studied by industry scholars. Additionally, these studies focus on optimizing the target through intelligent offloading or adaptive configuration. The authors in [30] observed ROI changes from the perspective of

the UE, when they decoupled the rendering and offloading parts with fast object tracking used locally in order to solve this problem. The authors in [31] investigated edge-end cloud collaboration for real-time video analytics and designed an online algorithm to achieve near-optimal utility by adjusting the quality of video frames generated on the UEs. The authors in [32] designed a new video configuration decision-making system, which examined the influence of video content on the frame rate and the resolution of the video stream. The authors in [33] optimized the video configuration and network bandwidth resource allocation using the Lyapunov and the Markov approximation, which solved the problems of resource limitation and network dynamic changes in edge-based video analysis systems. To consider fairness and long-term system cost, as well as optimizing the overall user QoE, the authors in [34] proposed an intelligent edge cache system to solve the bandwidth requirement and delay tolerance of 3600 panoramic video footage. To realize secure video sharing in vehicular edge computing, the authors in [35] designed an attribute-based encryption algorithm with static and dynamic attributes, and utilized a blockchain to record access strategies, which could ensure the data security and privacy of the video footage. In order to improve the QoE of live-streaming video, the authors in [36] first selected the candidate transcoding tasks by their contribution to popularity-weighted video quality and assigned these tasks to MEC in a greedy manner. The authors in [37] proposed a segment prefetching and edge caching algorithm to improve the QoE of Hyper Text Transfer Protocol (HTTP) adaptive video streaming. They first proposed and analyzed different segmentation prefetch strategies to dynamically adapt to the current conditions of the network and the needs of service providers. Moreover, they presented segment prefetching policies based on different approaches and techniques, and they studied their performance and feasibility. However, for real surveillance video, the resolution can only be selected downward. When the network resources are sufficient, we can then consider selecting the higher resolution video using super-resolution techniques to obtain higher video frame resolution and video analysis accuracy.

### 2.3. Video Offloading Based on DRL

In the last few years, the development of artificial intelligence (AI) technologies has led to rapid progress of DRL in modeling, routing, and resource management with a model-free environment. An adaptive video configuration network was pursued in [38] based on a black-box approach, independent of a detailed analytical performance model. The authors presented and designed an intelligent system named Cuttlefish, a type of smart coder which can adapt to the needs of the users without using any pre-programmed models or specific assumptions. The authors in [39] dealt with the problem of joint configuration adaptation and bandwidth allocation in an edge-assisted real-time video analysis system. They presented a novel approach which could select the configurations for multiple video streams immediately based on the state of the network and the content of the video. To work out the collaboration in the MEC network, an AI-based task allocation algorithm was presented in [40], which was trained by using a self-play strategy. The algorithm could detect a change in the network environment and adjust the resource allocation decision simultaneously. The authors in [41] proposed a two-layer learning model based on a DQN and a back propagation neural network to solve the joint decision of task offloading, wireless channel allocation, and image compression ratio selection in video analysis, and balance the accuracy of image recognition and processing delay. The authors in [42] presented a new approach to allocating resources in MEC networks using a radio map and DRL. Then, they presented a collaborative offloading and resource allocation algorithm which was used to solve the problem of reducing system latency and energy consumption. The authors in [43] considered the real-time video analytics of cameras based on edge coordination. In order to realize highly energy-efficient video analysis in a digital twin, a mobile device and edge coordination video analysis framework based on deep reinforcement learning was proposed, which takes energy consumption, analysis accuracy, and delay into consideration. However, using a DQN network to train the multi-parameter

problem necessitates the calculation of the probabilities of various actions and the selection of the action with the maximum probability, which leads to a high training delay and reduces effectiveness. Therefore, it is worthwhile to study how the training efficiency can be increased, and the precision of the network can be guaranteed.

## 3. System Model

The video edge scheduling model consists of two layers with different functionalities, the end layer and the edge layer, as shown in Figure 1. In the end layer, a set of UEs is randomly distributed on the ground, which can be expressed as M = {1,2, . . . ,$M$}. N = {1,2, . . . ,$N$} represents a set of clusters of NOMA, and K = {1,2, . . . ,$K$} denotes a set of subchannels. All UEs in a NOMA cluster share one subchannel at the same time for offloading, with each subchannel having an equal bandwidth. The UEs adopt a binary offloading rule, that is, the task of each UE must be processed locally or offloaded to the edge layer. The UEs continuously transmit video analysis tasks to the edge layer. In the edge layer, a MEC server is integrated on a computation access point (CAP), which receives the tasks offloaded to the edge layer, and the MEC server provides computing services for tasks.

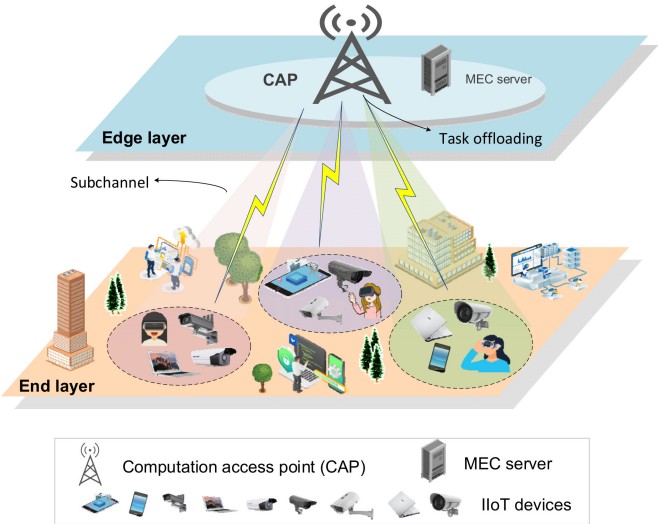

**Figure 1.** The video edge scheduling architecture.

The video stream calculation task of the UE $m$ is expressed as $D_m = \{l_m, c_m, T_m^{max}\}$, where $l_m$ represents the data size of the video stream task $D_m$, $c_m$ represents the total central processing unit (CPU) cycles required for the video stream task $D_m$, and $T_m^{max}$ denotes the maximum tolerance time for task $D_m$. After $T_m^{max}$, task $D_m$ will be declared the process ended in failure. The data size $l_m$ of task $D_m$ can be expressed as

$$l_m = \tau \cdot \eta_m^2, \tag{1}$$

where $\tau$ represents the number of bits required for a pixel to carry information. $\eta_m^2$ indicates the video frame resolution of task $D_m$.

Although the edge environment is constantly changing, the network state and video data are stable within a short time range, therefore we split the time into discrete time slots and each time slot has a duration. At the start of each time slot, the resources are reconfigured according to the present state and historical trend to obtain the best resource distribution status for the overall and long-term results. In the rest of this section, we explain the NOMA-enabled transmission model (Section 3.1) and the edge computing model (Section 3.2).

### 3.1. NOMA-Enabled Transmission Model

We assume that each UE can only be grouped into one NOMA cluster [44]. We define $x_{m,n} = 1$ to indicate that UE $m$ is assigned to the NOMA cluster $n$. On the other hand, $x_{m,n} = 0$ indicates that this assignment does not occur. Then,

$$\sum_{n=1}^{N} x_{m,n} = \begin{cases} 1, & \text{task } D_m \text{ transmit to ES through NOMA cluster } n, \\ 0, & \text{task } D_m \text{ is computed locally.} \end{cases} \tag{2}$$

Since the result obtained after video processing is very small, we do not consider the process of sending the result back to the UEs, and only consider the process of offloading the video stream to the MEC server. The uplink transmission rate of the UEs for the NOMA scheme is

$$R_m = \frac{1}{K} W \sum_{n=1}^{N} x_{m,n} \log_2 \left( 1 + \frac{p_{m,k} h_{m,k}}{\sigma_k^2 + \sum_{j \neq m} p_{j,k} h_{j,k}} \right), \tag{3}$$

where $W$ denotes the total transmission bandwidth, which may be bisected by $K$ subchannels; $p_{m,k}$ represents the transmit power of the UE $m$ to the MEC server on subchannel $k$; $h_{m,k}$ represents the channel gain between the UE $m$ and the MEC server on subchannel $k$; $\sigma_k^2$ indicates the noise power on the subchannel $k$. Then, the translation delay of UE $m$ can be expressed as

$$t_m^U = \frac{l_m}{R_m} \cdot \frac{1}{\rho_m}, \tag{4}$$

where $\rho_m$ is the compression ratio of the video frame for UE $m$, which is determined by video resolution and bit rate [45].

When the video stream task $D_m$ is offloaded to the edge for computing, the energy consumption generated by the UE $m$ during transmission is

$$e_m^U = p_{m,k} \cdot t_m^U. \tag{5}$$

### 3.2. Edge Computation Model

(1) *Edge Computing:* We denote $F$ as the total computing capacity of the MEC server, while $\kappa_m$ is the ratio of computing capacity allocated by the MEC server to the UE $m$. Thus, the computation delay of the task $D_m$ offloaded to the MEC server is

$$t_m^C = \frac{c_m}{\kappa_m F}. \tag{6}$$

The energy consumption generated by the UE $m$ during edge processing is

$$e_m^C = v \cdot (\kappa_m F)^2 \cdot c_m, \tag{7}$$

where $v = 10^{-27}$ is the effective switched capacitance of the CPU, determined by the CPU hardware architecture.

(2) *Local Computing:* For task $D_m$ computed locally, we use $F_m^{loc}$ to present the computing capacity of UE $m$. Thus, the latency of the task $D_m$ processed locally is

$$t_m^{loc} = \frac{c_m}{F_m^{loc}}, \tag{8}$$

Thus, the energy consumption of the task processed locally is

$$e_m^{loc} = v \cdot \left( F_m^{loc} \right)^2 \cdot c_m, \tag{9}$$

### 3.3. Problem Formulation

In order to optimize multiple conflicting goals equally, a common approach is to give different weights to these conflicting goals, and then to weigh and sum the goals. In this article, improving user accuracy and reducing processing latency are the basic goals. According to the reference [46], the analytical accuracy $\varphi_m$ of task $D_m$ is expressed as the ratio of the number of objects that are correctly identified to the total number of objects in a video frame, which can be expressed as

$$\varphi_m = 1 - 1.578e^{-6.5 \times 10^{-3}\eta_m}, \tag{10}$$

which is widely used in the relevant references [47–49].

Combined with Equations (4) and (6), the total latency generated by UE $m$ offloaded to the MEC server is composed of the transmission delay and computation delay, which can be expressed as

$$t_m^{off} = t_m^U + t_m^C = \frac{l_m}{R_m} \cdot \frac{1}{\gamma_m} + \frac{c_m}{\kappa_m F}, \tag{11}$$

Combined with Equations (5) and (7), the total energy consumption generated by UE $m$ offloaded to the MEC sever is composed of transmission energy consumption and computation energy consumption, which can be expressed as

$$e_m^{off} = e_m^U + e_m^C = p_{m,k} \cdot t_m^U + v \cdot (\kappa_m F)^2 \cdot c_m, \tag{12}$$

According to the assigned calculation model and communication model, the total latency required for task $D_m$ to be processed at $t$ is expressed as

$$T_m = \left(1 - \sum_{n=1}^{N} x_{m,n}\right) t_m^{loc} + \sum_{n=1}^{N} x_{m,n} t_m^{off}, \tag{13}$$

At the same time, the total energy consumption of the task $D_m$ at $t$ can be expressed as

$$E_m = \left(1 - \sum_{n=1}^{N} x_{m,n}\right) e_m^{loc} + \sum_{n=1}^{N} x_{m,n} e_m^{off}, \tag{14}$$

The states and video content are constantly changing, causing the offloading decision and resource allocation strategies to need to be constantly adjusted to accommodate the dynamics of our environment. When designing adaptive algorithms, our goal is to optimize the cost function, consisting of delay $T_m$, energy consumption $E_m$, and video analysis accuracy $\varphi_m$, under long-term resource constraints. Based on the design of the utility function in [47], the cost minimization function can be modeled as

$$
\begin{aligned}
\min \quad & \frac{1}{m}\sum_{m=1}^{M}\left[\omega_t T_m + \omega_e E_m - (1 - \omega_t - \omega_e)\varphi_m\right]\\
s.\,t. \quad & C_1 : x_{m,n} \in \{0,1\}, \forall m \in \mathcal{M}, n \in \mathcal{N}\\
& C_2 : \sum_{n=1}^{N} x_{m,n} \le 1, \forall m \in \mathcal{M}, n \in \mathcal{N}\\
& C_3 : T_m \le T_m^{\max}, E_m \le E_m^{\max}, \forall m \in \mathcal{M}\\
& C_4 : 0 \le p_{m,k} \le P_m^{\max}, \forall m \in \mathcal{M}, k \in \mathcal{K}\\
& C_5 : \sum_{m=1}^{M} \kappa_m \le 1, \forall m \in \mathcal{M}\\
& C_6 : \eta_m \ge \eta_m^{\min}, \forall m \in \mathcal{M}
\end{aligned} \tag{15}
$$

In the cost function, $\omega_t$ is the weight for the delay, and $\omega_e$ is the weight for the energy consumption, where $\omega_t + \omega_e < 1$. Constraint $C_1$ in the objective function guarantees that a UE can only be assigned to a NOMA cluster. Constraint $C_2$ denotes that the UEs can only select computed locally or offloaded to the MEC server. Constraint $C_3$ ensures that the

maximum total delay of UE $m$ must be less than the tolerance time $T_m^{max}$ of task $D_m$, and the total energy consumption of UE $m$ cannot exceed the threshold $E_m^{max}$. Constraint $C_4$ guarantees that the transmit power of UE $m$ cannot exceed the threshold $P_m^{max}$. Constraint $C_5$ ensures that the allocated computation capacity cannot exceed the total capacity of the MEC server. Constraint $C_6$ guarantees that the minimum video frame resolution of task $D_m$ must be higher than the threshold $\eta_m^{\min}$.

Two important challenges to solving this problem are the difficulty of the problem itself, and the prediction of future network status, video content, and other information. Since edge nodes typically run for months or years, in order to deal with the problem of unpredictable future information, it is necessary to relax the constraints in each time slot of the objective function to the average over a long period of time. In addition, the optimization problem is a mixed integer nonlinear program which is hard to resolve even if the future information is known. To address these two challenges, we need to design an algorithm that provides the best offloading and resource allocation for video streaming without being able to foresee future information.

## 4. Deep Reinforcement Learning-Based Algorithm

Based on the optimization target and constraints, the DRL-based algorithm is adopted. In the rest of this article, we first define the state space, the action space, and the reward function. Secondly, we present a more detailed description of the participant critic algorithm framework.

### 4.1. Deep Reinforcement Learning Model

The deep reinforcement learning process reformulates the computational offloading problem as a Markov Decision Process (MDP) model. A typical MDP model consists of a tuple $\{S, A, P, R, \gamma\}$ with five elements, where $S$ represents the state space, $A$ represents the finite action space, $P$ is the state transfer probability, $R$ represents the reward function, and $\gamma \in [0, 1]$ is the discount factor for future rewards. Each element of the MDP model tuple corresponds to the following meaning.

#### 4.1.1. State Space

At time slot $t$, the state of the UEs includes basic information about the computational task. The state space $S_{m,t} \in S_t$ can be expressed as

$$S_{m,t} = \left\{ s_{m,t} \middle| s_{m,t} = (l_m, \eta_m^2, h_{m,k}, T_m^{\max}) \right\}, \tag{16}$$

where $S_{m,t}$ denotes the state space at time slot $t$.

#### 4.1.2. Action Space

At time slot $t$, the action of UE $i$ is represented as

$$A_{m,t} = \{ a_{m,t} | a_{m,t} = (\alpha_m, \beta_m) \}, \tag{17}$$

It consists of two vectors: the task offloading and resource allocation vector $\alpha_m$, and the video frame resolution scale vector $\beta_m$. Vector $\alpha_m$ contains two actions: the resource allocation action $x_{m,n}$, and the offloading decision action $\sum_{n=1}^{N} x_{m,n}$. $x_{m,n}$ represents whether UE $m$ is assigned to the NOMA cluster $n$, and $\sum_{n=1}^{N} x_{m,n}$ represents whether task $D_m$ needs to be offloaded to the MEC server. Vector $\beta_m$ represents the action for the video frame resolution compression ratio selection, where $\beta_m \in [0.5, 1.5]$.

In the MEC system network proposed in this paper, the MEC server distributes the offloading and resource allocation policy to the UEs, however, the selection of the video frames resolution should also be determined.

4.1.3. Reward Function

The cost minimization function in this article contains multiple factors. Specifically, our goal is to reduce latency and energy consumption, as well as improve video analysis accuracy under long-term resource constraints. Therefore, the reward functions can be designed based on the optimization problem.

We propose JVFRS-TO-RA-DQN, which contains two DQN networks. The first DQN network selects the optimal offloading and resource allocation strategy, and the second DQN network selects the appropriate video frame resolution scaling factor to ensure the maximum accuracy of video analysis and to reduce the system delay and energy consumption. A detailed description of the two reward functions is given below.

After performing the action $A_{m,t}$, a reward $r_{m,t}$ will be obtained for the action $A_{m,t+1}$ that the edge server chooses to perform. The reward function is generally related to an objective function, which aims to minimize the delay of the system in the context of task offloading and resource allocation. However, the aim of reinforcement learning training is to obtain the maximum long-term accumulation of rewards. Thus, the offloading reward function of the UEs at time $t$ can be designed as

$$\xi_{m,t} = -(\omega_1 T_m + \omega_2 E_m). \tag{18}$$

For UEs, it should be penalized if the accuracy of the next state is not within the threshold after taking action $\alpha_{m,t}$. Therefore, the resolution scaling reward function is designed as

$$\zeta_{m,t} = (\omega_t + \omega_e - 1)\varphi_m, \tag{19}$$

Finally, we use $r_{m,t} = \xi_{m,t} + \zeta_{m,t}$ to represent the total reward of the system. By maximizing the long-term cumulative reward $r_{m,t}$, an efficient joint video frame resolution, task offloading, and resource allocation strategy, which we abbreviate as JVFRS-CO-RA-DQN in this paper, can be developed to achieve the minimization of system delay and energy consumption while improving video analysis accuracy.

*4.2. JVFRS-TO-RA-DQN Algorithm*

In this section, we propose JVFRS-TO-RA-DQN to solve the problem of joint video frame resolution scaling, task offloading, and resource allocation. The proposed algorithm is based on DQN, which can study the offline historical data through the experience of a simulation without requiring full environmental knowledge. The detailed algorithm is shown in Figure 2.

According to the state of the system at present, the DQN algorithm maximizes the predefined reward function by choosing an $A_{m,t}$ from a limited sum of actions.

In the process of training, apart from state $S_{m,t}$, action $A_{m,t}$, policy $\pi$, and reward function $r_{m,t}$, the state-action value function $Q^\pi(S_{m,t}, A_{m,t})$ determines the action $A_{m,t}$ of the state $S_{m,t}$ through a mapping function $\pi(A_{m,t} | S_{m,t})$. If $Q^\pi(S_{m,t}, A_{m,t})$ is updated at each time step, then it is assumed that it will converge to the optimum state-action value function $Q^{\pi'}(S_{m,t}, A_{m,t})$. According to the Bellman equation, the evaluation of the quality of a specific action in a given specific state can be expressed as

$$Q^\pi(S_{m,t}, A_{m,t}) = (1-\lambda)Q^\pi(S_{m,t}, A_{m,t}) + \lambda(r_{m,t} + \gamma \max_{A'} Q(S_{m,t+1}, A')), \tag{20}$$

where $\lambda$ represents the learning rate which reflects the rate of the algorithm adapting to a new environment, where $\lambda \in (0, 1]$. Since the complexity of Equation (20) is exponentially related to the number of state-action pairs, it is more difficult to solve $Q$ values when the state-action pairs increase. In order to accurately calculate the $Q$ values, predicting the values of $Q$ between different state-action pairs is significant, and also represents the hinge of the DQN algorithm.

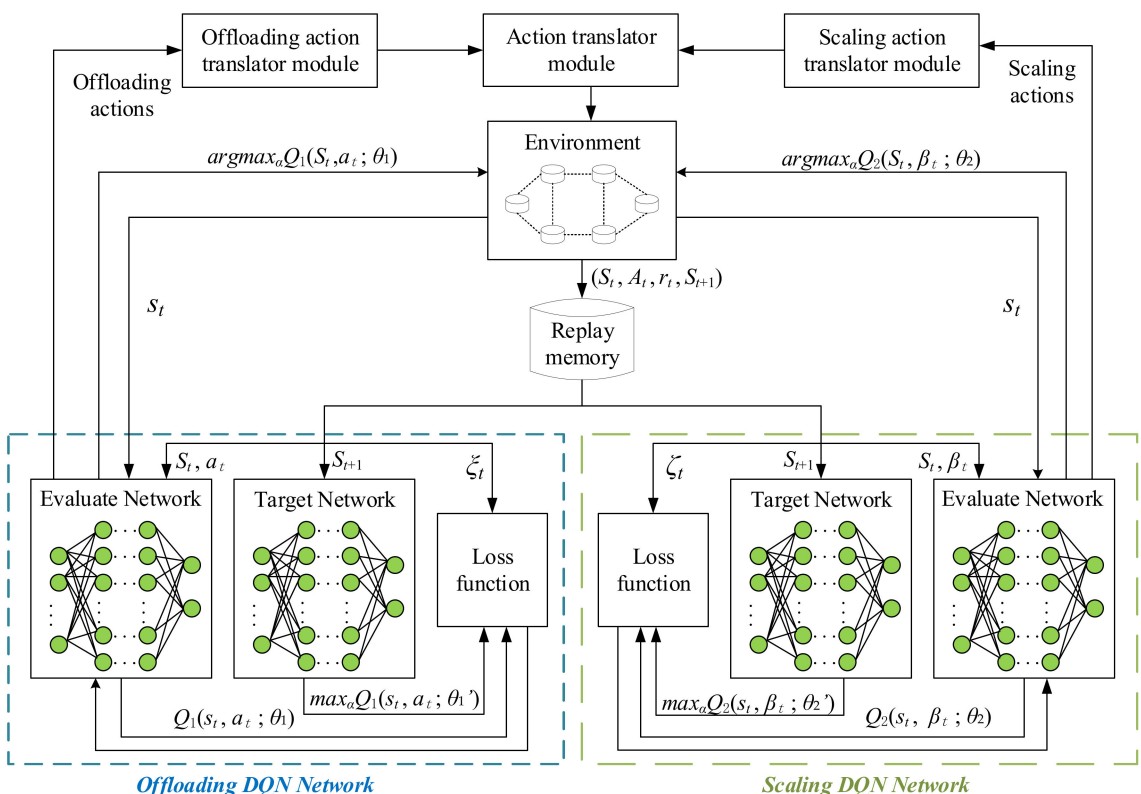

**Figure 2.** An illustration of the JVFRS-TO-RA-DQN architecture.

In contrast to using traditional tabular Q-learning for prediction, DQN has a special replay memory structure to store the data generated after each step, including every step. When the network is in training, it extracts some memory from the replay memory for experiential learning. The replay memory has enough training data to fit the $Q$ values of different state-action pairs, which leads to

$$Q^\pi(S, A) \approx \hat{Q}(S, A, \theta), \tag{21}$$

where $\hat{Q}(\cdot; \theta)$ is a deep learning network function denoted by $\theta$. $Q^\pi(S, A)$ is updated by minimizing the loss function, which is defined as

$$L(\theta) = \mathbb{F}\left[\left(r_{m,t} + \gamma\max_{A'}Q(S_{m,t+1}, A') - Q(S, A, \theta)\right)^2\right], \tag{22}$$

The gradient descent algorithm is utilized to minimize the loss in Equation (22) and therefore to update the weight $\theta$, so as to make it possible to minimize the error between the evaluation and the target. As a result, the neural network can predict more accurately as the training process continues. The JVFRS-CO-RA-DQN algorithm is explained in Algorithm 1.

---

**Algorithm 1**: JVFRS-CO-RA-DQN algorithm

**Input**: $D_m$, $w$, $F$, $\gamma$.
**Output:** $\alpha_m$, $\beta_m$.

---

1:   Initialize the evaluate network with random weights as $\theta$
2:   Initialize the target networks as a copy of the evaluate network with random weights as $\theta'$
3:   Initialize replay memory $D$
4:   Initialize an empty state set $S\_Set$
5:   **for** episode = 1 to *Max* **do**
6:       Initialize state $S_{m,t}$ in Equation (16)
7:       **for** $t < T$ **do**
8:          With probability $\varepsilon$ to select a random offloading and resource allocation decision $\alpha_{m,t}$; with probability $\delta$ to select a random resolution $\beta_{m,t}$
9:          Execute action $\alpha_{m,t}$, receive a reward $\xi_{m,t}$; execute action $\beta_{m,t}$, receive a reward $\zeta_{m,t}$
10:        Combine $\alpha_{m,t}$ and $\beta_{m,t}$ as $A_{m,t}$, calculate $r_{m,t}$ with $\xi_{m,t}$ and $\zeta_{m,t}$, and observe the next state $S_{m,t+1}$
11:        Store interaction tuple $\{S_{m,t}, A_{m,t}, r_{m,t}, S_{m,t+1}\}$ in $D$
12:        Sample a random tuple $\{S_{m,t}, A_{m,t}, r_{m,t}, S_{m,t+1}\}$ from $D$
13:        Compute the offloading target $Q$ value and the scaling target $Q$ value
14:        Train the offloading target $Q$ value and the scaling target $Q$ value
15:        Perform gradient descent with respect to $\theta$
16:        Update the evaluate Q-network and target Q-network
17:       **end for**
18: **end for**

---

## 5. Experimental Results and Discussion

In this section, we evaluate the performance of JVFRS-CO-RA-DQN in terms of the cost under different network conditions, the delay and analysis accuracy under different minimum frame resolutions, and the convergence performance. We first describe the parameter settings before delving into the simulation results.

### 5.1. Parameter Settings

In this study, we adapt Python 3.7 as the software tool to simulate the framework, and the deep learning framework in JVFRS-CO-RA-DQN is PyTorch 1.4.0. The hardware is a computer with Intel I7-13700HQ @ 2.5 GHz and 16-GB of memory. To verify the effectiveness of the JVFRS-CO-RA-DQN algorithm proposed in this study, a network consisting of one MEC, four NOMA clusters, and ten UEs is considered for experiments, with the UEs randomly distributed within [0, 200] m from the MEC. The total communication bandwidth $W$ is 12 MHz. We define $c_m$ in correlation with the data size of the task as $c_m = c_m^{bit} l_m$, where $c_m^{bit}$ is 100 cycles/bit. The computational capacity of the MEC $F$ is 12 GHz, the computational capacity of the UEs. $F_m^{loc}$ is [0.4, 2] GHz, the average energy consumption threshold $E_m^{max}$ is 15 J, and the number of bits required to carry information per unit pixel of video $\tau$ is 24. The minimum transmission power $p$ of the UEs is 0.5 W, while the maximum tolerance time $t_m^{max}$ of task $D_m$ is 30 ms. Based on the experimental data from the reference [46], we adopt Equation (10) as the analytic accuracy function on both edge servers and the UEs, and the video frame resolution is higher than 40,000 px (200 × 200). The parameters are shown in Table 1.

**Table 1.** Simulation parameters.

| Parameters | Value |
|---|---|
| Number of UEs, $M$ | 10 |
| Number of NOMA clusters, $k$ | 4 |
| The distance between the MEC server and UEs | [0, 200] m |
| The total communication bandwidth, $W$ | 12 MHz |
| CPU cycles required for unit bit task, $c_m^{bit}$ | 100 cycles/bit |
| The computational capacity of the MEC server, $F$ | 10 GHz |
| The computational capacity of the UEs, $F_m^{loc}$ | [0.4, 2] GHz |
| Average energy consumption threshold, $E_m^{max}$ | 15 J |
| Required bits representing one pixel, $\tau$ | 24 |
| Maximum transmission power, $p$ | 0.5 W |
| Maximum tolerance time for task, $t_m^{max}$ | 30 ms |
| Minimum video frame resolution, $\eta_m^{min}$ | 40,000 px (200 × 200) |
| Constant of the IoT device, $v$ | $1 \times 10^{-27}$ |
| Compression ratio of the video frame for UE, $\rho_m$ | 74 |
| Discount factor, $\gamma$ | [0, 1] |
| Batch size, $Z$ | 128 |
| Replay buffer, $B$ | 100 |

*5.2. Result Analysis*

To assess its performance fairly, the proposed scheme was compared with four baseline schemes:

(1) Local Computing Only (LCO): the video streams are processed totally at the UEs with $\sum^{n=1 \ N} x_{m,n} = 0, \forall m \in M$, which has a fixed video frame resolution.

(2) Edge Computing Only via OMA (ECO-OMA): the video streams are totally offloaded to and processed at the MEC server with $x_{m,n} = 1, \forall m \in M, n \in N$, which has a fixed video frame resolution.

(3) JVFRS-TO-RA-DQN via OMA (JVFRS-TO-RA-DQN-OMA): Unlike JVFRS-TO-RA-DQN, task $D_m$ generated by UE $m$ are offloaded to the MEC server through OMA. Each UE has an independent subchannel. We use $y_m$ to denote whether task $D_m$ offloaded to the MEC server, $y_m = 1$ denotes that task $D_m$ were offloaded to MEC sever; otherwise, $y_m = 0$.

(4) Task offloading and a resource allocation algorithm based on DQN via NOMA (TO-RA-DQN-NOMA) [50]: Compared with JVFRS-TO-RA-DQN, TO-RA-DQN-NOMA does not consider the change in video frame resolution, which means that it has a fixed video frame resolution.

(5) Maximum accuracy algorithm via NOMA (MA-NOMA) [46]: Compared with JVFRS-TO-RA-DQN, MA-NOMA implements maximum accuracy with the largest frame resolutions in NOMA.

A comparison between the six algorithms is shown in Table 2.

**Table 2.** Algorithm comparison.

| | 0–1 Offloading | NOMA | Resolution | Delay and Energy |
|---|---|---|---|---|
| LCO | × | × | × | √ |
| ECO-OMA | × | × | × | √ |
| JVFRS-TO-RA-DQN-OMA | √ | × | √ | √ |
| TO-RA-DQN-NOMA | √ | √ | × | √ |
| MA-NOMA | √ | √ | √ | × |
| JVFRS-TO-RA-DQN-NOMA | √ | √ | √ | √ |

We first take the experiment in a specific scene with $k = 4$, $M = 10$, $F = 10$ GHz, $W = 12$ MHz and $\eta_m^{min} = 200 \times 200$ px. Then, we record the system delay under five

different schemes. We take 1500 experiments and average the experimental data, which is exhibited in Table 3. The average latency of the JVFRS-TO-RA-DQN scheme is 167.71 ms. That is about 48.14% less than the JVFRS-TO-RA-DQN-OMA scheme, about 33.38% less than the TO-RA-DQN-NOMA scheme, and about 59.01% less than the MA-NOMA scheme.

**Table 3.** The average latency of six schemes.

| | LCO | ECO-OMA | JVFRS-TO-RA-DQN-OMA | TO-RA-DQN-NOMA | MA-NOMA | Proposed |
|---|---|---|---|---|---|---|
| The bandwidth of subchannel (MHz) | 1.2 | 1.2 | 1.2 | 3 | 3 | 3 |
| Average delay (ms) | 644.54 | 801.49 | 323.42 | 251.77 | 409.66 | 167.71 |

The influence of different communication bandwidths on the cost function is shown in Figure 3. We take 1500 experiments and average the experimental data in a scene with $k = 4$, $M = 10$, $F = 10$ GHz, $\eta_m^{min} = 200 \times 200$ px and $W$ ranging from 2 MHz to 12 MHz. Figure 3 exhibits the effect of different communication bandwidths on the cost in this scenario under six schemes. First of all, the cost of all schemes decreases as the communication bandwidth increases, except for that of the LCO scheme. This is because the LCO scheme transmits only at the UEs, and a change in the network communication bandwidth does not affect it. Additionally, the average cost of LCO schemes almost does not change. However, the cost of other schemes decreases as the communication bandwidth increases because every UE is able to allocate more bandwidth and the delay and energy consumption of communication transmission is also reduced. The cost of the proposed algorithm is reduced by about 49.51% compared with the JVFRS-TO-RA-DQN-OMA scheme at $W = 6$ MHz, and by about 34.17% compared with the TO-RA-DQN-NOMA scheme at $W = 6$ MHz.

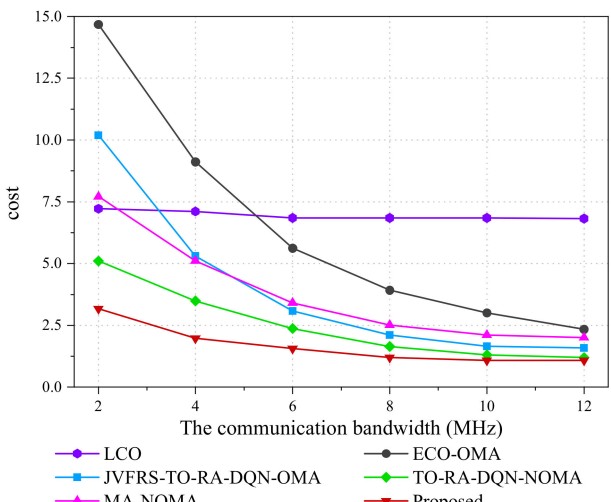

**Figure 3.** The effect of communication bandwidth on the cost.

Figure 4 depicts the effect of the different computational capacities of the MEC server on the cost function. We take 1500 experiments and average the experimental data in a scenario with $k = 4$, $M = 10$, $W = 10$ GHz, $\eta_m^{min} = 200 \times 200$ px and $F$ ranging from 2 MHz to 12 MHz. Due to the fact that the UEs do not utilize the computing resources of the MEC server, the LCO schemes will not change as the computational capacity of the MEC server increases. However, the other schemes are reduced as the computational capacity of the MEC server increases since more computer source is allocated for MEC, with the computation time also being shortened accordingly. It is obvious that the average cost is mostly affected by other elements when the computational capacity of the MEC server is much bigger than the computational capacity of the UEs.

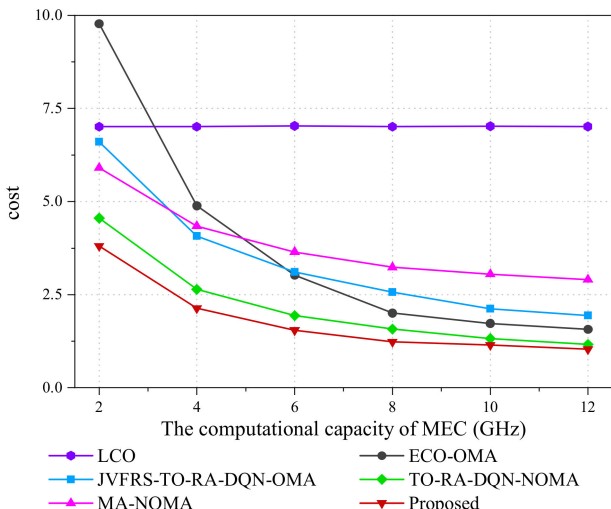

**Figure 4.** The effect of computational capacity of MEC on the cost.

Except for the delay and energy consumption, the video analytic accuracy is also affected by the video frame resolution. We modify the minimum video frame resolutions in this experiment to estimate the influence of resolution on accuracy. We take 1500 experiments and average the experimental data in a scenario with $k = 4$, $M = 10$, $F = 10$ GHz, $W = 10$ GHz, and $\eta_m^{min}$ ranging from $200 \times 200$ px to $700 \times 700$ px. When calculating the accuracy, we assume that the system can detect all objects when the video frame resolution is $700 \times 700$ px. The video frame resolution is lower than $700 \times 700$ px, and the optimized resolution is between the minimum and the maximum video frame resolutions. The video frame resolution of the LCO scheme, the ECO-OMA scheme, and the TO-RA-DQN-NOMA scheme is identified as the average of the minimum and the maximum video frame resolutions.

Figure 5 shows the effects of different minimum frame resolutions on the average delay. As illustrated in Figure 5, the average delay of all the algorithms increases as the minimum video frame resolutions change, except for the MA-NOMA. This is because the MA-NOMA always keeps the highest resolution, and the change of minimum frame resolution does not affect it. The proposed algorithm maintains minimum average delay, which means the proposed algorithm can adjust the resolution of video frame adaptively to reduce the system average delay. The average delay of the JVFRS-TO-RA-DQN-OMA scheme is higher than that of the proposed algorithm, which means that the delay of the NOMA enabled MEC system is superior to that of the OMA.

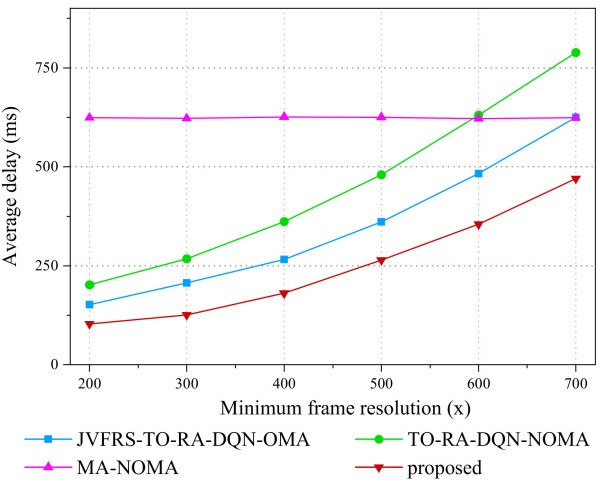

**Figure 5.** The effect of minimum frame resolution on average delay.

Figure 6 shows the effects of different minimum frame resolutions on video analytic accuracy. It is demonstrated that the MA-NOMA algorithm always maintains the highest level of accuracy, which is because the system can detect all objects when the video frame resolution is 700 × 700 px. The analytic accuracy of the proposed algorithm is lower than TO-RA-DQN-NOMA's, for the proposed algorithm sacrifices the analytic accuracy to reduce delay and energy consumption. The influence of delay and energy on the system is gradually increasing when the minimum video frame resolution is larger than 500 × 500 px, the video analytic accuracy is almost steady.

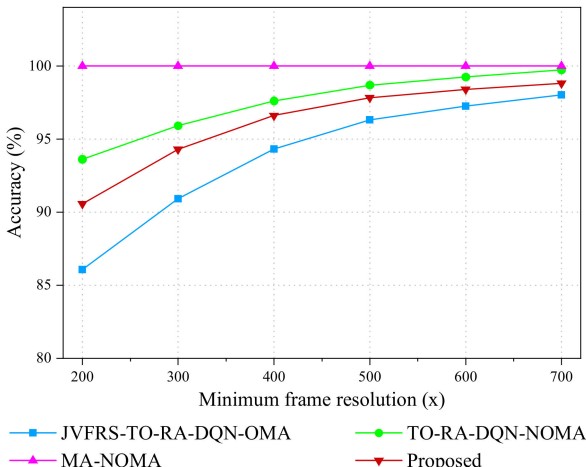

**Figure 6.** The effect of minimum frame resolution on video analytic accuracy.

We then depict the process of convergence with the TO-RA-DQN-NOMA algorithm and the proposed algorithm in a scenario with $k = 4$, $M = 10$, $F = 10$ GHz, $W = 12$ MHz and $\eta_m^{min} = 200 \times 200$ px. Figure 7 depicts the performance differences between the TO-RA-DQN-NOMA algorithm and the proposed algorithm at different learning rates. On the one hand, the proposed algorithm obtains correspondingly higher rewards than the TO-RA-DQN-NOMA scheme. This is owing to the fact that different algorithms lead to different offloading decisions, which means that the video frames offloaded to MEC are different, while the average delay and video analysis accuracy are also different. Compared with TO-RA-DQN-NOMA algorithm, the simulation data in Table 4 indicates that the proposed algorithm has better performance than the TO-RA-DQN-NOMA algorithm, in terms of video analytic accuracy. Moreover, the convergence rate of the proposed algorithm is higher than that of the TO-RA-DQN-NOMA algorithm. Furthermore, it was discovered that after training about 200 epochs, the learning rates of the proposed algorithm are $10^{-6}$ and $10^{-7}$, converging to a reward value. On the other hand, after training more than 400 epochs, the learning rates of the TO-RA-DQN-NOMA algorithm are $10^{-6}$ and $10^{-7}$, converging to a reward value.

**Table 4.** The video analytic accuracy of the two schemes.

|  | TO-RA-DQN-NOMA | Proposed |
|---|---|---|
| Learning rate of $10^{-6}$ | 95.87% | 98.74% |
| Learning rate of $10^{-7}$ | 95.96% | 98.82% |

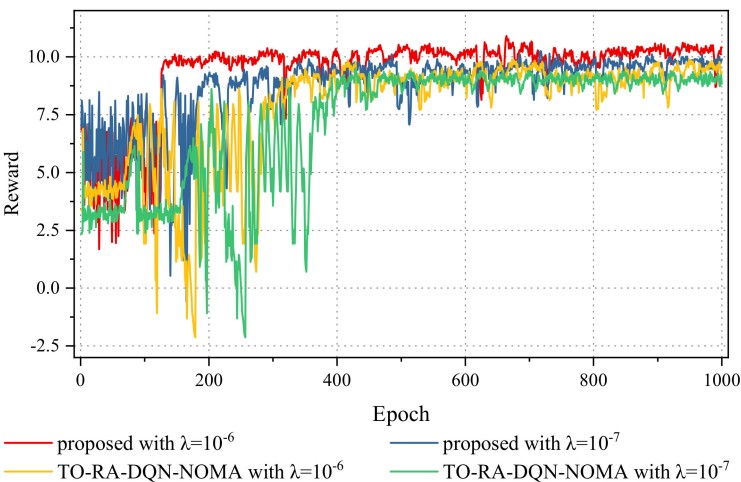

**Figure 7.** Convergence performance of the proposed algorithm and the TO−RA−DQN−NOMA algorithm under different learning rates.

Figure 8 shows the convergence performance of the proposed algorithm under different numbers of UEs. We take the experiment in a scene with $k = 4$, $F = 10$ GHz, $W = 10$ GHz, $\eta_m^{min} = 200 \times 200$ px, $M = 10, 15$, and 20. The algorithm converges rapidly and steadily no matter how many UEs there are. Furthermore, the average reward of 10 UEs apparently exceeds that of 15 UEs. The average reward of 10 UEs converges to a reward for training at approximately 120 epochs, the average reward of 15 UEs converges to a reward for training at approximately 270 epochs, and the average reward of 20 UEs converges to a reward value at over 400 epochs. This is due to the fact that more UEs are able to offload computing tasks at a higher efficiency than when there are fewer UEs, which reduces the energy consumption of UEs while enhancing the users' QoE on the basis of latency, energy consumption, and video analytic accuracy.

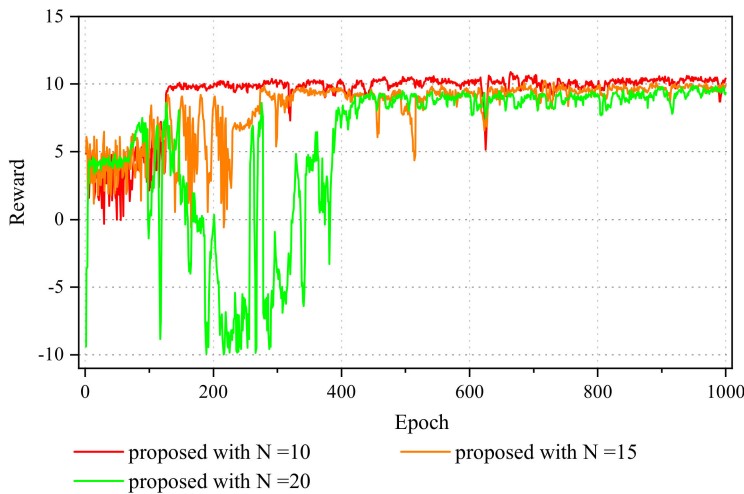

**Figure 8.** Convergence performance of the proposed algorithm with different numbers of UEs.

## 6. Conclusions

With the popularization of video surveillance applications and the diversification of functional applications, real-time video stream analysis is of great value for intelligent monitoring, smart cities, autonomous driving, and other scenarios. In this paper, we designed a NOMA-enabled smart video analysis system with multiple UEs for the purpose of improving the video analytic accuracy, reducing the average delay, and decreasing the energy consumption. Aiming to optimize the QoE of UEs, we formulated a cost minimization problem composed of delay, energy, and accuracy to weigh up the relationship between

these three parameters. The cost minimum function was an NP-hard problem with high dimensional nonlinear mixed integer programming, which was difficult to calculate the optimal solution. The JVFRS-TO-RA-DQN algorithm was proposed to solve the above problem. The proposed algorithm contains two DQN networks one was used to select the offloading and resource allocation actions, and the other was used to select video frame resolution scaling actions, which effectively overcame the sparsity of the single-layer reward function and accelerated the training convergence speed. A large number of simulation experiments showed that the JVFRS-TO-RA-DQN algorithm can achieve better performance in improving video analysis accuracy, reducing total delay and decreasing energy consumption compared to other baseline schemes.

**Author Contributions:** Conceptualization, S.G. and J.Z.; methodology, S.G.; software, S.G.; validation, S.G.; formal analysis, S.G. and Y.W.; investigation, S.G. and N.F.; resources, S.G.; data curation, S.G.; writing—original draft preparation, S.G. and N.F.; writing—review and editing, S.G. and Y.W.; visualization, S.G. and Y.W.; supervision, Z.W.; project administration, Z.W.; funding acquisition, J.Z. All authors have read and agreed to the published version of the manuscript.

**Funding:** This research was funded by the Science and Technology Research Project of Higher Education Institutions of Hebei Province under grant No. QN2020193 and No. ZD2020171, Hebei Province Innovation Capability Improvement Plan Program (22567624H), the Handan Science and Technology Research and Development Program under grant No. 21422031288, and the Provincial Innovation Funding Project for Graduate Students of Hebei Province under grant No. CXZZSS2023120.

**Institutional Review Board Statement:** Not applicable.

**Informed Consent Statement:** Not applicable.

**Data Availability Statement:** The data in this paper is not publicly available at this time.

**Conflicts of Interest:** The authors declare no conflict of interest.

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
