# Peer review of "Deep Reinforcement Learning-Based Video Offloading and Resource Allocation in NOMA-Enabled Networks"

_futureinternet, doi:10.3390/fi15050184_

Round 1

Reviewer 1 Report

The paper presents an algorithm for video edge offloading optimize and resource allocation decision for an edge computing environment. The authors propose a three-layer NOMA-assisted video edge scheduling architecture aiming to improve the efficiency of data offloading and a JVFRS-TO-RA-DQN algorithm to solve the optimization problem targeting the analysis accuracy, the delay, and the energy consumption.

The paper is well structured, and the algorithm is interesting and proves good performance. The related work is clearly explained. The algorithm is clearly explained, and many useful details are included. However, there are some issues that have to be addressed by the authors, as following.

1. Abbreviations must be defined at the first usage (e.g., MEC [49]).

2. The authors must explain better the overall system architecture, e.g., on which layer the Algorithm1  is executing?

3. All the parameters of eq. (16) must be explained.

4. In Figures 3 and 4 it is not clear how the cost is evaluated. More specifically, how the cost function is defined?

5. In Figure 6 it is not clear what video analytics accuracy the authors are referring to. What application is targeted by the experimental setup and how this is implemented? The testing results presented here are very confusing. It is completely unclear to which experiments refer the authors under the name “video analytic”. The accuracy of such systems depends on many other factors than resolution as types of analytic, relations between the dataset used for training and the running environment etc. To avoid confusion and interpretation of relevance, standard data sets are used for such measurements and comparisons. The suggestion for the authors is to add more information related to the experimental environment, data used for testing, etc.

6. The caption of Figure 8 is inaccurate (“number of” is missing) and should be corrected to “Convergence performance of the proposed algorithm under different number of IoT devices.”

1. Some parts of the document (especially the introduction and the related work) contain many long and confusing sentences and must be rewritten. E.g., [69] “However, it is not easy to analyze real-time video offloading to the edge for analysis is also time-consuming and resource-intensive, which needs to require dynamic trade-offs between edge computing and video frame transmission.”

2. There is a large number of small errors and typos that need to be corrected. E.g. [35] “Low frame rate (1.25 Hz) moving image sequences generate more than 100M data per second.” – It is not clear if the authors refer to Mb (bits) or MB (bytes)? ; Capital letter for the beginning of a phrase [170] “analytical performance modeling. the authors presented and designed an intelligent system …”, [433] “shortened accordingly. it is obvious that the average cost …”, [505] “integer programming problem. we proposed the …” etc.).

Author Response

JIJUN ZHAO Research group

School of Information and Electrical Engineering

Hebei University of Engineering

14 May 2023

Dear Reviewer 1

Future Internet

Thank you for your valuable suggestion and the Reviewer's comments concerning our manuscript entitled “Deep Reinforcement Learning-Based Video Offloading and Resource Allocation in NOMA-Enabled Networks” (Manuscript ID: futureinternet-2383077). The paper has been improved a lot. We have revised the manuscript according to the recommendations and have provided detailed responses to each comment below. For the reviewers’ convenience, we have highlighted major changes in the revised manuscript in yellow background.

Reply to Reviewer's comments:

Reviewer 1:

Comments and Suggestions for Authors

  1. Abbreviations must be defined at the first usage (e.g., MEC [49]).

Re: Thank you very much for your careful check.

We mistakenly believed that a single abbreviation in the abstract is enough. All the abbreviations at the first usage are defined in the manuscript:

Line 42: user equipments (UEs)

Line 49: mobile edge computing (MEC)

Line 59: non-orthogonal multiple access (NOMA)

Line 65: quality of experience (QoE)

Line 78: deep reinforcement learning (DRL)

Line 93-94: joint video frame resolution scaling, task offloading, and resource allocation algorithm based on DQN algorithm (JVFRS-TO-RA-DQN)

Line 140: unmanned aerial vehicle (UAV)

Line 235: central processing unit (CPU)

  1. The authors must explain better the overall system architecture, e.g., on which layer the Algorithm1 is executing?

Re: Thank you very much for your valuable suggestion.

The original overall system architecture was divided into three layers according to the data generation, data collection and data computing. But it could not show the process of the video tasks generated by the terminal device transferring to the edge server well. We pointed the system architecture again, as shown below.

We rewrote this section on line 220-230 as:

The video edge scheduling model consists of two layers with different functionalities, the end layer and the edge layer, as shown in Figure 1. In the end layer, a set of UEs is randomly distributed on the ground, which can be expressed as M={1,2,…,M}. N={1,2,…,N} represents the set of the clusters of NOMA, and K={1,2,…,K} denotes the set of subchannels. All UEs in a NOMA cluster share one subchannel at the same time for offloading, with each subchannel has an equal bandwidth. The UEs adopt a binary offloading rule, that is, the task of each UE must be processed locally or offloaded to edge layer. The UEs continuously transmit video analysis tasks to the edge layer. In the edge layer, a MEC server integrated on a computation access point (CAP), which receives the tasks offloaded to the edge layer, and the MEC server provide computing services for tasks.

What’s more, when a computing task is generated on UE, Algorithm 1 determines whether the task needs to be offloaded based on the current network state. The UE sends the computing task to the CAP if the task needs to be offloaded, then the MEC server integrated on the CAP provides computing services for the task. Otherwise, the task will be processed locally.

  1. All the parameters of eq. (16) must be explained.

Re: Thank you very much for your valuable suggestion.

The eq. (16) expresses the state space Sm,t, and all the explanation of parameters have been mentioned above eq. (16). The parameters are explained as follows:

At time slot t, the state of the UEs includes basic information about the computational task. The Sm,t in our manuscript consists of four parameters, lm, η2 m, hm,k, and Tmax m:

lm represents the data size of the input video stream task Dm, which is mentioned in line 234.

η2 m indicates the video frame resolution of the task Dm, which is mentioned in line 238-239.

hm,k represents the channel gain between the UE m and ES on subchannel k, which is mentioned in line 257-258.

Tmax m denotes the maximum tolerance time for task Dm, which is mentioned in line 235-236.

Since these parameters are explained when they first appear in the manuscript, we do not explain them again after eq. (16).

  1. In Figures 3 and 4 it is not clear how the cost is evaluated. More specifically, how the cost function is defined?

Re: Thank you very much for your valuable suggestion.

The cost function in manuscript is defined by the weighted sum of delay, energy consumption and video analysis accuracy, which is defined in line 294-297. The equation of delay, energy consumption and video analysis accuracy are respectively introduced in eq.(13), eq.(14) and eq.(10). In order to easy understanding, We added the corresponding symbols after the three parameters. We formulated the cost function as eq. (15), which is also the optimization function.

We modified line 294-297 to mention the problem more clearly:

When designing adaptive algorithms, our goal is to optimize the cost function consisting of delay Tm, energy consumption Em, and video analysis accuracy φm under long-term resource constraints. Based on the design of the utility function in [47], the cost minimization function can be modeled as

  1. In Figure 6 it is not clear what video analytics accuracy the authors are referring to. What application is targeted by the experimental setup and how this is implemented? The testing results presented here are very confusing. It is completely unclear to which experiments refer the authors under the name “video analytic”. The accuracy of such systems depends on many other factors than resolution as types of analytic, relations between the dataset used for training and the running environment etc. To avoid confusion and interpretation of relevance, standard data sets are used for such measurements and comparisons. The suggestion for the authors is to add more information related to the experimental environment, data used for testing, etc.

Re: Thank you so much for your valuable suggestion.

Our manuscript only considers the impact of video frame resolution on the video analysis accuracy. The video analysis accuracy is influenced by many factors, such as video frame resolution, frame rate, bit rate, the number of subjects in the dataset and the speed subjects move, etc. So, it is very difficult to fit a general video analysis accuracy function. The manuscript only studies the impact of resolution on the video analysis accuracy, and uses eq.(10) to denote the relationship between video analysis accuracy and video frame resolution, which is fitted from the experimental data in the reference [46] (line 275-280). The authors in [46] implemented two object recognition algorithms, YOLO and SSD, to verify the relationship between video frame resolution and video analysis accuracy. The hardware in [46] is a computer with Nvidia Quadro M4000 GPU. They defined the analytics accuracy as the ratio between the number of correctly recognized objects and that of total objects in a video frame. Based on the observations, they used eq.(10) to define the relationship between analysis accuracy and video frame resolution.

Many references [47-49] relate to offloading and resource allocation also use eq.(10) to represent the relationship between analysis accuracy and video frame resolution.

In addition, we adopt Python 3.7 as the software tool to simulate the framework in this manuscript. The deep learning framework in JVFRS-CO-RA-DQN is PyTorch 1.4.0. The hardware is a computer with Intel I7-13700HQ @ 2.5GHz and 16-GB memory. The initial environment of the experiment is shown in Table 1. In each time slot, the network will automatically generate a computing task, which will include information, such as video duration, the initial resolution , generated from which terminal device, and tolerable delay. The network will determine how to process generated tasks. Although the tasks are randomly generated, we have taken 1500 experiments and averaged the experimental data. Therefore, the experimental settings are reasonable, and the experimental results can reflect the performance of JVFRS-CO-RA-DQN in reducing offloading delay, energy consumption and improving video analysis accuracy.

In order to more accurately describe the experimental environment, we added the corresponding experimental environment before each experimental analysis:

Line 409-411: We adapt Python 3.7 as the software tool to simulate the framework in this manuscript, and the deep learning framework in JVFRS-CO-RA-DQN is PyTorch 1.4.0. The hardware is a computer with Intel I7-13700HQ @ 2.5GHz and 16-GB memory.

Line 447-448: We first take the experiment in a specific scene with k=4, M=10, F=10GHz, W=12MHz and ηmin m=200×200 pixels.

Line 456-457: We take 1500 experiments and average the experimental data in a scene with k=4, M=10, F=10GHz, ηmin m=200×200 pixels and W range from 2MHz to 12MHz.

Line 472-474: Figure 4 depicts the effect of the different computational capacities of the MEC sever on the cost function. We take 1500 experiments and average the experimental data in the scenario with k=4, M=10, W=10GHz, ηmin m=200×200 pixels and F range from 2MHz to 12MHz.

Line 486-488: We take 1500 experiments and average the experimental data in the scenario with k=4, M=10, F=10GHz, W=10GHz, ηmin m range from 200×200 pixels to 700×700 pixels.

Line 519-520: We then depict the process of convergence with TO-RA-DQN-NOMA algorithm and the proposed algorithm in the scenario with k=4, M=10, F=10GHz, W=12MHz and ηmin m=200×200 pixels.

Line 536-537: We take the experiment in a scene with k=4, F=10GHz, W=10GHz, ηmin m=200×200 pixels, M=10, 15, and 20.

  1. The caption of Figure 8 is inaccurate (“number of” is missing) and should be corrected to “Convergence performance of the proposed algorithm under different number of IoT devices.”

Re: Thank you so much for your careful check. We changed the caption of Figure 8 to: Convergence performance of the proposed algorithm under different numbers of UEs.

Comments on the Quality of English Language

  1. Some parts of the document (especially the introduction and the related work) contain many long and confusing sentences and must be rewritten. E.g., [69] “However, it is not easy to analyze real-time video offloading to the edge for analysis is also time-consuming and resource-intensive, which needs to require dynamic trade-offs between edge computing and video frame transmission.”

Re: Thank you so much for your careful check. We rewrote this passage on line 67-69 as:

However, real-time video analysis is not easy. Offloading video to the edge for analysis requires high demands on resources and latency, which needs to dynamically balance the transmission and computation processes.

According to the reviewer’s good comment, we went through the whole paper carefully to rewritten the sentence which is too long and confusing. On this basis, we also standardized the language through the MDPI English polishing service. We hope that the revised manuscript will be more clear in expressions. Thanks again for your valuable suggestion.

  1. There is a large number of small errors and typos that need to be corrected. E.g. [35] “Low frame rate (1.25 Hz) moving image sequences generate more than 100M data per second.” – It is not clear if the authors refer to Mb (bits) or MB (bytes)? ; Capital letter for the beginning of a phrase [170] “analytical performance modeling. the authors presented and designed an intelligent system …”, [433] “shortened accordingly. it is obvious that the average cost …”, [505] “integer programming problem. we proposed the …” etc.).

Re: Thank you so much for your careful check.

We rewrote the passage on line 36-37 as:

Low frame rate (1.25 Hz) moving image sequences generate more than 100 Mb data per second.

We rewrote the passage on line 193-195 as:

The authors presented and designed an intelligent system named Cuttlefish, a type of smart coder which can adapt to the needs of the users without using any pre-programmed models or specific assumptions.

We rewrote the passage on line 478-480 as:

It is obvious that the average cost is mostly affected by other elements when the computational capacity of the MEC server is much bigger than the computational capacity of the UEs.

We rewrote the passage on line 559-560 as:

The JVFRS-TO-RA-DQN algorithm was proposed to solve the above problem.

What’s more, according to the reviewer’s comment, we went through the whole manuscript carefully to check the sentence mistakes and other issues, such as prepositions, singular and plural, determiner, pronoun, conjunction, and the verb form that related to the linguistic presentation. Besides, the spelling, sentence structure, typos, and spaces in this manuscript had been thoroughly proofread, and the mistakes had been corrected accordingly. We hope that the revised manuscript will be more accurate in expressions. Thanks again for your valuable suggestion.

We have carefully improved the manuscript based on the reviewer's suggestions and have made some modifications to the manuscript. These modifications will not affect the framework of this paper. We sincerely thank the editors and reviewers for their enthusiastic work and hope that the modifications will be approved. Once again, we thank the reviewers for their comments and suggestions.

Yours sincerely

Siyu Gao, Yuchen Wang, Nan Feng, Zhongcheng Wei, and Jijun Zhao

Reviewer 2 Report

In this paper, the authors propose a non-orthogonal multiple access (NOMA) based edge real-time video analysis framework for improving the quality of experience (QoE) of users in video surveillance systems. The proposed framework includes one edge server (ES) and multiple user equipments (UEs), and a cost minimization problem composed of delay, energy, and accuracy is formulated to achieve high accuracy and low latency video stream analysis. To efficiently solve this problem, the authors propose a joint video frame resolution scaling, task offloading, and resource allocation algorithm based on the Deep Q-Learning Network (JVFRS-TO-RA-DQN).

The JVFRS-TO-RA-DQN algorithm consists of two DQN networks that respectively select the offloading and resource allocation action, and the resolution scaling action. The proposed algorithm effectively overcomes the sparsity of the single-layer reward function and accelerates the training convergence speed. Experimental results show that JVFRS-TO-RA-DQN can effectively reduce the cost of transmission and computation and outperforms other baseline algorithms in terms of convergence speed.

Overall, this paper provides a valuable contribution to the field of video surveillance systems by proposing a NOMA-based edge real-time video analysis framework and a joint optimization algorithm that can improve QoE for users. The experimental results demonstrate the effectiveness of the proposed algorithm, which could have potential applications in other areas where real-time video analysis is critical.

·       Title: It is hard to understand the current title: Can the authors rewrite the title as "Improving Video Offloading and Resource Allocation in NOMA Networks Using Deep Reinforcement Learning"

·       System Model: The motivation for the system model is missing. To answer this, the authors need to specify the research goals and what they want to measure clearly.

·       Problem Formulation: Authors wrote on the problem formulation. There is a need to mention the problem statement after finalizing the formulation.

·       The titles of section 5 and 5.2 are same. Please mention the goals of the experimental setup.

·       What are the limitations of the study and the future work.

Author Response

JIJUN ZHAO Research group

School of Information and Electrical Engineering

Hebei University of Engineering

14 March 2023

Dear Reviewer 2

Future Internet

Thank you for your valuable suggestion and the Reviewer's comments concerning our manuscript entitled “Deep Reinforcement Learning-Based Video Offloading and Resource
Allocation in NOMA-Enabled Networks” (Manuscript ID: futureinternet-2383077). The paper has been improved a lot. We have revised the manuscript according to the recommendations and have provided detailed responses to each comment below. For the reviewers’ convenience, we have highlighted major changes in the revised manuscript in yellow background.

Reply to Reviewer's comments:

Reviewer 2:

Comments and Suggestions for Authors

  1. Title: It is hard to understand the current title: Can the authors rewrite the title as "Improving Video Offloading and Resource Allocation in NOMA Networks Using Deep Reinforcement Learning"

Re: Thank you very much for your valuable suggestion.

The proposed algorithm in the manuscript is a new algorithm based on DQN. It is also a deep reinforcement learning based algorithm. So, “Deep Reinforcement Learning Based” would be more appropriate to describe the meaning in this manuscript. DQN is a value-based deep reinforcement learning algorithm which uses two neural networks with the same structure and different parameters. The difference is that one is used for training and the other will not be trained in the short term. By adopting the untrained network, it can ensure that the “target Q value” remains stable at least in the short term. The proposed algorithm in this paper uses two DQN networks to train offloading actions and video frame resolution selection actions, which overcomes the sparsity of the single-layer reward function and accelerates the training convergence speed.

  1. System Model: The motivation for the system model is missing. To answer this, the authors need to specify the research goals and what they want to measure clearly.

Re: Thank you very much for your valuable suggestion.

The motivation for the system model is to reduce the delay and energy consumption of video offloading, and improve the accuracy of video analysis. 

A cost minimization function consisting of delay Tm, energy consumption Em, and video analysis accuracy φm was proposed to optimize three parameters, which is also the objective function. To calculate the delay and energy consumption of video offloading, the manuscript divided the offloading process into NOMA-enabled transmission process and computing process. If the system determines that the task needs to be offloaded to the MEC, the total delay can be expressed as the sum of transmission delay and computing delay. Similarly, the total energy consumption can be expressed as the sum of transmission energy consumption and computing energy consumption. Otherwise, the task will be processed locally on the terminal device without transmission process. The total delay can be expressed as the local computation delay. Similarly, the total energy consumption can be expressed as the local computing energy consumption.

The video analysis accuracy depends on many factors as resolution, types of analytic, the number of users, video frame rate, the speed at which users move, relations between the dataset used for training and the running environment etc. So, it is very difficult to fit a general video analysis accuracy function. The manuscript only studies the impact of resolution on the accuracy of video analysis, and uses eq.(10) to denote the relationship between video analysis accuracy and video frame resolution, which is fitted from the experimental data in the reference “An Edge Network Orchestrator for Mobile Augmented Reality”. We normalized the delay, energy consumption and video analysis accuracy, and the cost function is the average of M UEs’ cost.

Finally, we verified the changes of the cost function under different network environments, the effects of different initial minimization frame resolutions on delay and accuracy, and the convergence performance of rewards under different network environments.

  1. Problem Formulation: Authors wrote on the problem formulation. There is a need to mention the problem statement after finalizing the formulation.

Re: Thank you very much for your valuable suggestion.

According to the reviewer’s good comment, we modified line 294-297 to mention the problem more clearly:

When designing adaptive algorithms, our goal is to optimize the cost function consisting of delay Tm, energy consumption Em, and video analysis accuracy φm under long-term resource constraints. Based on the design of the utility function in [47], the cost minimization function can be modeled as

  1. The titles of section 5 and 5.2 are same. Please mention the goals of the experimental setup.

Re: Thank you very much for your valuable suggestion.

The main content of section 5 is experiment and result analysis. The proposed algorithm in section 4 is simulated and analyzed in python. Next, the performance of the proposed algorithm and the baseline algorithm is compared to prove the effectiveness of the proposed algorithm. Among them,section 5.1 is the parameter setting, which describes the initial environment and parameter values of the simulation; 5.2 is the analysis of the simulation results. we verify the changes of the cost function under different network environments, the effects of different initial minimization frame resolutions on delay and accuracy, and the convergence performance of rewards under different network environments.

The titles of section 5 and 5.2 should not be the same, in order to avoid any misunderstanding, we changed the line 411-415 as:

  1. Experimentand Results Discussion

In this section, we evaluate the performance of JVFRS-CO-RA-DQN in terms of the cost under different network conditions, the delay and analysis accuracy under different minimum frame resolution, and the convergence performance. We first describe the parameter settings before delving into the simulation results.

  1. What are the limitations of the study and the future work.

Re: Thank you very much for your valuable suggestion. The limitations of the study mainly includes two aspects:

  • The manuscriptonly considers the impact of video frame resolution on the data size and the video analysis accuracy. In fact, the change in video frame rate, frame resolution and bit rate may all affect the data size and the video analysis accuracy. Therefore, how to consider the effects of three parameters jointly on the video offloading is the focus of the further research.
  • The manuscriptmainly studies the task offloading and resource allocation problems in MEC, the proposed algorithm is simulated and analyzed in Python without using real video datasets. Due to the large differences between different real datasets, the algorithm proposed in this manuscript may need to be further improved. The advance work will focus on the video offloading algorithm under real datasets.

We have carefully improved the manuscript based on the reviewer's suggestions and have made some modifications to the manuscript. These modifications will not affect the framework of this paper. We sincerely thank the editors and reviewers for their enthusiastic work and hope that the modifications will be approved. Once again, we thank the reviewers for their comments and suggestions.

Yours sincerely

Siyu Gao, Yuchen Wang, Nan Feng, Zhongcheng Wei, and Jijun Zhao

Reviewer 3 Report

- QoE - Quality of Experience is not explained at first mention. Check other abbreviations, such as JVFRS, DQN .... It seems that many abbreviations are not explained at first mention in the text.

- Did you take into account privacy concerns and risk of cyber attacks? 

- You could improve your references by adding a few from 2023, but these from 2022 are relevant and satisfactory.

- Application in autonomous driving  is not well explained. Is it possible to use it in real time reliably? Traffic applications mean that the situation is fast changing. Latency could cause  traffic accidents. Your latency tests are based on simulation, not a real world. You should improve this part of the paper. 

- Rewrite contributions to enhance language and style issues. Lines 103-114.

- Line 317: formulaic

- Proof read the manuscript.

- Improve language, because it degrades feeling about the paper.

Author Response

JIJUN ZHAO Research group

School of Information and Electrical Engineering

Hebei University of Engineering

14 March 2023

Dear Reviewer 3

Future Internet

Thank you for your valuable suggestion and the Reviewer's comments concerning our manuscript entitled “Deep Reinforcement Learning-Based Video Offloading and Resource
Allocation in NOMA-Enabled Networks” (Manuscript ID: futureinternet-2383077). The paper has been improved a lot. We have revised the manuscript according to the recommendations and have provided detailed responses to each comment below. For the reviewers’ convenience, we have highlighted major changes in the revised manuscript in yellow background.

Reply to Reviewer's comments:

Reviewer 3:

Comments and Suggestions for Authors

  1. QoE - Quality of Experience is not explained at first mention. Check other abbreviations, such as JVFRS, DQN .... It seems that many abbreviations are not explained at first mention in the text.

Re: Thank you very much for your careful check.

We mistakenly believed that a single abbreviation in the abstract is enough. All the abbreviations at the first usage were defined in the manuscript:

Line 42: user equipments (UEs)

Line 49: mobile edge computing (MEC)

Line 59: non-orthogonal multiple access (NOMA)

Line 65: quality of experience (QoE)

Line 78: deep reinforcement learning (DRL)

Line 93-94: joint video frame resolution scaling, task offloading, and resource allocation algorithm based on DQN algorithm (JVFRS-TO-RA-DQN)

Line 140: unmanned aerial vehicle (UAV)

Line 235: central processing unit (CPU)

  1. Did you take into account privacy concerns and risk of cyber attacks?

Re: Thank you very much for your valuable suggestion.

Privacy concerns and risk of cyber attacks were not considered in our manuscript, even if they were widely concerned. We mainly paid attention to the video edge offloading and allocation problem, the proposed algorithm reduced the latency and energy consumption, as well as improved video analysis accuracy of the offloading process by optimizing the offloading decision and resolution scaling action. A network environment with less interference from other factors can effectively control the variables and better reflect the performance advantages of the proposed algorithm. We also take into account the privacy concerns and risk of cyber attacks in the network, but the privacy concerns of the network is our next research direction, which will be reflected in our future work.

  1. You could improve your references by adding a few from 2023, but these from 2022 are relevant and satisfactory.

Re: Thank you very much for your valuable suggestion. As the reviewer recommended, we carefully checked the related works section in this manuscript, and accordingly updated the related paper with 5 references from 2023 were added. Besides, we elaborated on the detail and novelty of each paper. The Related Works are as follows:

Line169-183: To consider fairness and long-term system cost, as well as optimizing the overall user QoE, the authors in [34] proposed an intelligent edge cache system to solve the bandwidth requirement and delay tolerance of 3600 panoramic video footage. To realize secure video sharing in vehicular edge computing, the authors in [35] designed an attribute-based encryption algorithm with static and dynamic attributes, and utilized a blockchain to record access strategies, which could ensure the data security and privacy of the video footage. In order to improve the QoE of live-streaming video, the authors in [36] first selected candidate transcoding tasks by their contribution to popularity-weighted video quality and assigned these tasks to MEC in a greedy manner. The authors in [37] proposed a segment prefetching and edge caching algorithm to improve the QoE of Hyper Text Transfer Protocol (HTTP) adaptive video streaming. They first proposed and analyzed different segmentation prefetch strategies to dynamically adapt to the current conditions of the network and the needs of service providers. Moreover, they presented segment prefetching policies based on different approaches and techniques, and they studied their performance and feasibility.

Line209-214: The authors in [43] considered the real-time video analytic of cameras based on edge coordination. In order to realize highly energy-efficient video analysis in digital twin, a mobile device and edge coordination video analysis framework based on deep reinforcement learning was proposed, which take energy consumption, analysis accuracy and delay into consideration.

The updated references are given as follows:

-In References

  1. Jin Y, Liu J, Wang F, et al. Ebublio: Edge assisted multi-user 360-degree video streaming[J]. IEEE Internet of Things Journal, 2023.
  2. Jiang B, He Q, Liu P, et al. Blockchain Empowered Secure Video Sharing With Access Control for Vehicular Edge Computing[J]. IEEE Transactions on Intelligent Transportation Systems, 2023.
  3. Lee D, Kim Y, Song M. Cost-Effective, Quality-Oriented Transcoding of Live-Streamed Video on Edge-Servers[J]. IEEE Transactions on Services Computing, 2023.
  4. Aguilar-Armijo J, Timmerer C, Hellwagner H. SPACE: Segment Prefetching and Caching at the Edge for Adaptive Video Streaming[J]. IEEE Access, 2023, 11: 21783-21798.
  5. Yang P, Hou J, Yu L, et al. Edge-coordinated energy-efficient video analytics for digital twin in 6G[J]. China Communications, 2023, 20(2): 14-25.
  6. Application in autonomous driving is not well explained. Is it possible to use it in real time reliably? Traffic applications mean that the situation is fast changing. Latency could cause traffic accidents. Your latency tests are based on simulation, not a real world. You should improve this part of the paper.

Re: Thank you very much for your valuable suggestion. Real-time video analysis technology has important value in scenarios such as intelligent surveillance, smart city, autonomous driving and so on. But this manuscript is not based on the autonomous driving scenario. The video analysis accuracy depends on many other factors than resolution as types of analytic, the number of users, video frame rate, the speed at which users move, relations between the dataset used for training and the running environment etc. So, it is very difficult to fit a general video analysis accuracy function.

The manuscript only considers the impact of video frame resolution on the video analysis accuracy. The manuscript uses eq.(10) to denote the relationship between video analysis accuracy and video frame resolution, which was fitted from the experimental data in the reference “An Edge Network Orchestrator for Mobile Augmented Reality”. It is obvious from eq.(1) and eq.(10) that the higher video frame resolution is, the higher video analysis accuracy is, and the larger data size of the video stream task is, which results in extra delay and energy consumption of the offloading process. Hence, the proposed algorithm needs to determine select which resolution scaling action and offloading decision to adopt for the video task can enhance the reward, which means, to minimize the cost function. The manuscript utilizes python simulation to manipulate the network variables, which demonstrates the performance advantages of the proposed algorithm.

Moreover, the manuscript considers the impact of simultaneous variations in video frame rate and video resolution on the network, and will explore the performance of the proposed algorithm with real video chip.

Comments on the Quality of English Language

  1. Rewrite contributions to enhance language and style issues. Lines 103-114.

Re: Thank you very much for your valuable suggestion. We deeply analyze and summarize the innovative contributions of the proposed algorithms, and add them to the introduction part of the revised manuscript.

We rewrote this passage on line 96-111 as:

  • A three-layer NOMA-assisted video edge scheduling architecture is proposed, where UEs are divided into different clusters of NOMA, and the tasks generated by UEs in the same cluster are offloaded over a common subchannel to improve the offloading efficiency.
  • An attempt is made to optimize the QoE of UE by formulating a cost-minimization problem composed of delay, energy, and accuracy in order to weigh up the relationship between these three parameters.
  • The JVFRS-TO-RA-DQN algorithm is proposed to solve the joint optimization problem. The JVFRS-TO-RA-DQN algorithm contains two DQN networks, one is used to select offloading and resource allocation action, and the other is used to select video frame resolution scaling action, which effectively overcomes the sparsity of the single-layer reward function and accelerates the training convergence speed.
  • The experimental results show that the JVFRS-TO-RA-DQN algorithm can achieve better performance gains in terms of improving video analysis accuracy, reducing total delay and decreasing energy consumption compared to other baseline schemes.
  1. Line 317: formulaic.

Re: Thank you so much for your careful check. To avoid misunderstanding, we rewrote this passage on line 345 as:

The cost minimization function in this article contains multiple factors.

  1. Proof read the manuscript.

Re: We thank the reviewer for the comment.

We went through the whole paper carefully to check the sentence mistakes and other issues, such as prepositions, singular and plural, determiner, pronoun, conjunction, and the verb form that related to the linguistic presentation. Besides, the spelling, sentence structure, typos, and spaces in this manuscript were thoroughly proofread, and the mistakes were corrected accordingly. Thanks again for your valuable suggestion.

  1. Improve language, because it degrades feeling about the paper.

Re: Thank you very much for your valuable suggestion. 

We read the manuscript again carefully and revised sentences which were too long or could cause misunderstanding. At the same time, we adjusted the sentence pattern, unified the tense, and enhanced the language's feeling about the paper. On this basis, we also standardized the language through the MDPI English polishing service. We hope that the revised manuscript will be more clear and more accurate in expressions. Thanks again for your valuable suggestion.

We have carefully improved the manuscript based on the reviewer's suggestions and have made some modifications to the manuscript. These modifications will not affect the framework of this paper. We sincerely thank the editors and reviewers for their enthusiastic work and hope that the modifications will be approved. Once again, we thank the reviewers for their comments and suggestions.

Yours sincerely

Siyu Gao, Yuchen Wang, Nan Feng, Zhongcheng Wei, and Jijun Zhao

Reviewer 4 Report

In this paper, the authors propose a NOMA-based method for real-time sharing of information (videos), optimizing energy and accuracy.

The idea seems promising, and it is also presented with a good approach; however, some questions should be answered:

Concerning the method, the authors exploit the proposed JVFRS-TO-RA-DQN algorithm and test its performance with some specific metrics. It sounds great, and in fact the methodology is correct; however, the only real comparison is with TO-RA-DQN-NOMA [39] since the other test is executed on JVFRS-TO-RA-DQN with OMA. Do you think it could be enough to prove the system's quality? Could it be more interesting to propose at least a comparison with another SOTA method?

The JVFRS-TO-RA-DQN is just expanded in the Abstract section. We suggest checking the acronyms and exploding them in the text at least the first time.

The experimental methodology is a bit confusing. A section is dedicated to both experimental setup and results; we encourage the authors to separate them and provide more details, particularly in the first one (no clear starting scenario is given).

The related works mainly focus on NOMAs, MEC Scenarios, and other IoT contexts. There is no consideration of the application areas, e.g., Robotics (even if [27] treat it, no details are provided), UAV, or healthcare (even if in [1] some examples are provided). Thus, we suggest integrating the references like the following ones:

Q. Cui, X. Zhao, W. Ni, Z. Hu, X. Tao and P. Zhang, "Multi-Agent Deep Reinforcement Learning-Based Interdependent Computing for Mobile Edge Computing-Assisted Robot Teams," in IEEE Transactions on Vehicular Technology, doi: 10.1109/TVT.2022.3232806.

Z. Yang et al., "AI-Driven UAV-NOMA-MEC in Next Generation Wireless Networks," in IEEE Wireless Communications, vol. 28, no. 5, pp. 66-73, October 2021, doi: 10.1109/MWC.121.2100058.

X. Liu, Y. Liu, Z. Yang, X. Yue, C. Wang and Y. Chen, "Integrated 3C in NOMA-Enabled Remote-E-Health Systems," in IEEE Wireless Communications, vol. 28, no. 3, pp. 62-68, June 2021, doi: 10.1109/MWC.001.2000403.

Some letters in formulas (lines 197 and 198) seem to be images... Is it true? We strongly suggest changing them to text format.

Even if no noticeable errors could be identified, some sentences are quite hard to read. We suggest improving the flow of the document.

Author Response

JIJUN ZHAO Research group

School of Information and Electrical Engineering

Hebei University of Engineering

14 March 2023

Dear Reviewer 4

Future Internet

Thank you for your valuable suggestion and the Reviewer's comments concerning our manuscript entitled “Deep Reinforcement Learning-Based Video Offloading and Resource
Allocation in NOMA-Enabled Networks” (Manuscript ID: futureinternet-2383077). The paper has been improved a lot. We have revised the manuscript according to the recommendations and have provided detailed responses to each comment below. For the reviewers’ convenience, we have highlighted major changes in the revised manuscript in yellow background.

Reply to Reviewer's comments:

Reviewer 4:

Comments and Suggestions for Authors

  1. Concerning the method, the authors exploit the proposed JVFRS-TO-RA-DQN algorithm and test its performance with some specific metrics. It sounds great, and in fact the methodology is correct; however, the only real comparison is with TO-RA-DQN-NOMA [39] since the other test is executed on JVFRS-TO-RA-DQN with OMA. Do you think it could be enough to prove the system's quality? Could it be more interesting to propose at least a comparison with another SOTA method?

Re: Thank you very much for your valuable suggestion.

In order to prove the performance of the JVFRS-TO-RA-DQN algorithm more strictly, we added a baseline algorithm, which implement maximum accuracy with the largest frame resolutions in NOMA (MA-NOMA)[46].

Then, the comparison between algorithms is shown in Table 2.

Table 2. Algorithm Comparison

0-1 offloading

NOMA

resolution

Delay and energy

LCO

×

×

×

ECO-OMA

×

×

×

JVFRS-TO-RA-DQN-OMA

×

TO-RA-DQN-NOMA

×

MA-NOMA

×

JVFRS-TO-RA-DQN-NOMA

We added part of the experiment and modified the experimental analysis in manuscript, the sentences in yellow background are the major changes.(line 426-514)

  1. The JVFRS-TO-RA-DQN is just expanded in the Abstract section. We suggest checking the acronyms and exploding them in the text at least the first time.

Re: Thank you very much for your careful check.

We mistakenly believed that a single abbreviation in the abstract is enough. All the abbreviations at the first usage were defined in the manuscript:

Line 42: user equipments (UEs)

Line 49: mobile edge computing (MEC)

Line 59: non-orthogonal multiple access (NOMA)

Line 65: quality of experience (QoE)

Line 78: deep reinforcement learning (DRL)

Line 93-94: joint video frame resolution scaling, task offloading, and resource allocation algorithm based on DQN algorithm (JVFRS-TO-RA-DQN)

Line 140: unmanned aerial vehicle (UAV)

Line 235: central processing unit (CPU)

  1. The experimental methodology is a bit confusing. A section is dedicated to both experimental setup and results; we encourage the authors to separate them and provide more details, particularly in the first one (no clear starting scenario is given).

Re: Thank you very much for your careful check.

In order to more accurately describe the experimental environment, we added the corresponding experimental environment before each experimental analysis:

Line 409-411: We adapt Python 3.7 as the software tool to simulate the framework in this manuscript, and the deep learning framework in JVFRS-CO-RA-DQN is PyTorch 1.4.0. The hardware is a computer with Intel I7-13700HQ @ 2.5GHz and 16-GB memory.

Line 447-448: We first take the experiment in a specific scene with k=4, M=10, F=10GHz, W=12MHz and ηmin m=200×200 pixels.

Line 456-457: We take 1500 experiments and average the experimental data in a scene with k=4, M=10, F=10GHz, ηmin m=200×200 pixels and W range from 2MHz to 12MHz.

Line 472-474: Figure 4 depicts the effect of the different computational capacities of the MEC sever on the cost function. We take 1500 experiments and average the experimental data in the scenario with k=4, M=10, W=10GHz, ηmin m=200×200 pixels and F range from 2MHz to 12MHz.

Line 486-488: We take 1500 experiments and average the experimental data in the scenario with k=4, M=10, F=10GHz, W=10GHz, ηmin m range from 200×200 pixels to 700×700 pixels.

Line 519-520: We then depict the process of convergence with TO-RA-DQN-NOMA algorithm and the proposed algorithm in the scenario with k=4, M=10, F=10GHz, W=12MHz and ηmin m=200×200 pixels.

Line 536-537: We take the experiment in a scene with k=4, F=10GHz, W=10GHz, ηmin m=200×200 pixels, M=10, 15, and 20.

  1. The related works mainly focus on NOMAs, MEC Scenarios, and other IoT contexts. There is no consideration of the application areas, e.g., Robotics (even if [27] treat it, no details are provided), UAV, or healthcare (even if in [1] some examples are provided). Thus, we suggest integrating the references like the following ones:

Re: Thank you very much for your careful check. Based on your suggestions, we summarized the application scenarios in related work:

(line139-154) In addition, NOMA technology is also applied in many practical situations to improve  spectrum utilization, such as robotics, unmanned aerial vehicle (UAV), and smart healthcare scenarios, e.g., where multiple users offload tasks to the MEC simultaneously. The authors in [26] proposed a communication enabled indoor intelligent robots (IRs) service framework, which adopted NOMA to support highly reliable communications. The efficiency and communication reliability of IRs was maximized using a DRL based algorithm. The authors in [27] considered a framework for computation offloading in which UAVs used NOMA and MEC techniques to serve mobile users. They introduced federated learning and reinforcement learning to solve the problem of privacy restriction between UAVs. In order to satisfy the ultra-reliable low-latency connectivity requirements of remote-e-Health systems, the authors in [28] considered applying NOMA to the e-Health systems, and proved that NOMA exhibits an excellent performance in the scenarios of fifth-generation and beyond. These solutions provide certain insights in applying NOMA to enable efficient task offloading in MEC scenarios efficiently, while also providing a feasible scheme through which to solve the efficient transmission of video data.

The updated references are given as follows:

-In References

  1. Cui Q, Zhao X, Ni W, et al. Multi-Agent Deep Reinforcement Learning-Based Interdependent Computing for Mobile Edge Computing-Assisted Robot Teams[J]. IEEE Transactions on Vehicular Technology, 2022.
  2. Zhong R, Liu X, Liu Y, et al. Path design and resource management for NOMA enhanced indoor intelligent robots[J]. IEEE Transactions on Wireless Communications, 2022, 21(10): 8007-8021.
  3. Yang Z, Chen M, Liu X, et al. AI-driven UAV-NOMA-MEC in next generation wireless networks[J]. IEEE Wireless Communications, 2021, 28(5): 66-73.
  4. Liu X, Liu Y, Yang Z, et al. Integrated 3c in noma-enabled remote-e-health systems[J]. IEEE Wireless Communications, 2021, 28(3): 62-68.
  5. Some letters in formulas (lines 197 and 198) seem to be images... Is it true? We strongly suggest changing them to text format.

Re: Thank you very much for your careful check. All the mathematical expressions in the manuscript are made by mathtype and they are not images. Because of the line spacing problem, it was not suitable to insert mathtype formulas in the text, so we replaced them with text forms.We rewrote this passage on line 223 as:

M={1,2,…,M}. N={1,2,…,N} represents a set of clusters of NOMA, and K={1,2,…,K} denotes

Comments on the Quality of English Language

  1. Even if no noticeable errors could be identified, some sentences are quite hard to read. We suggest improving the flow of the document.

Re: Thank you very much for your careful check. According to the reviewer’s good comment, we went through the whole paper carefully to revise the sentence which was hard to read. We checked the sentence mistakes and other issues, such as prepositions, singular and plural, determiner, pronoun, conjunction, and the verb form that related to the linguistic presentation. Besides, the spelling, sentence structure, typos, and spaces in this manuscript were been thoroughly proofread, and the mistakes were corrected accordingly. On this basis, we also standardized the language through the MDPI English polishing service. We hope that the revised manuscript will be more clear and more accurate in expressions. Thanks again for your valuable suggestion.

We have carefully improved the manuscript based on the reviewer's suggestions and have made some modifications to the manuscript. These modifications will not affect the framework of this paper. We sincerely thank the editors and reviewers for their enthusiastic work and hope that the modifications will be approved. Once again, we thank the reviewers for their comments and suggestions.

Yours sincerely

Siyu Gao, Yuchen Wang, Nan Feng, Zhongcheng Wei, and Jijun Zhao

Round 2

Reviewer 1 Report

The revised version of the manuscript adds some necessary clarifications in the description and validation of the proposed method and removes most of the reported errors. The authors succeed in removing inconsistencies in the figure’s title, adding the missing explanation of some equation parameters, detailing the cost function of the adaptive algorithm, and better describing the experimental environment. In conclusion, the general improvements of the manuscript allow me to recommend it for publication in the present form.